# Suicide as an incident of severe patient harm: a retrospective cohort study of investigations after suicide in Swedish healthcare in a 13-year perspective

Elin Fröding ,[1,2] Boel Andersson Gäre,[1,3] Åsa Westrin,[4,5] Axel Ros [1,2]

► Prepublication history and additional materials for this paper is available online. To view these files, please visit the journal online (http://dx.doi.org/10.1136/bmjopen-2020-044068).

[1]Jönköping University, Jönköping, Sweden
[2]Region Jönköpings län, Jönköping, Sweden
[3]Futurum, Region Jönköpings län, Jönköping, Sweden
[4]Division of Psychiatry, Department of Clinical Sciences, Lund University, Lund, Sweden
[5]Region Skåne, Psychiatry Research Skåne, Office for Psychiatry and Habilitation, Lund, Sweden

**Correspondence to**
Elin Fröding; elin.froding@rjl.se

## ABSTRACT

**Objectives** To explore how mandatory reporting to the supervisory authority of suicides among recipients of healthcare services has influenced associated investigations conducted by the healthcare services, the lessons obtained and whether any suicide-prevention-related improvements in terms of patient safety had followed.

**Design and settings** Retrospective study of reports from Swedish primary and secondary healthcare to the supervisory authority after suicide.

**Participants** Cohort 1: the cases reported to the supervisory authority in 2006, from the time the reporting of suicides became mandatory, to 2007 (n=279). Cohort 2: the cases reported in 2015, a period of well-established reporting (n=436). Cohort 3: the cases reported from September 2017, which was the time the law regarding reporting was removed, to November 2019 (n=316).

**Primary and secondary outcome measures** Demographic data and received treatment in the months preceding suicide were registered. Reported deficiencies in healthcare and actions were categorised by using a coding scheme, analysed per individual and aggregated per cohort. Separate notes were made when a deficiency or action was related to a healthcare-service routine.

**Results** The investigations largely adopted a microsystem perspective, focusing on final patient contact, throughout the overall study period. Updating existing or developing new routines as well as educational actions were increasingly proposed over time, while sharing conclusions across departments rarely was recommended.

**Conclusions** The mandatory reporting of suicides as potential cases of patient harm was shown to be restricted to information transfer between healthcare providers and the supervisory authority, rather than fostering participative improvement of patient safety for suicidal patients. The similarity in outcomes across the cohorts, regardless of changes in legislation, suggests that the investigations were adapted to suit the structure of the authority's reports rather than the specific incident type, and that no new service improvements or lessons are being identified.

## BACKGROUND

Deaths that occur as a result of patient harm represent a contrast to healthcare services'

### Strengths and limitations of this study

► To our knowledge, this is the first evaluation of the outcomes of investigations of specific types of patient harm over time, here exemplified by suicide.
► All investigations concerned the same kind of incident; suicides, and the data were population-based.
► All data were based on the healthcare providers' investigations and reports to the supervisory authority, the content in these reports is regulated by law; however, the quality of analysis differs, which was not evaluated in this study.
► All data collection and categorisation were conducted by only one researcher, which rendered categorisation vulnerable to bias; however this ensured a high level of consistency.

aim of a high level of patient safety, and such incidents can serve as powerful motivators for learning and improvement.[1 2] In recent decades, efforts to increase patient safety have been intensified. In particular, the reporting and investigating of cases of severe patient injury in order to identify risks and improve patient safety have become widespread safety-improvement strategies.[2] This reflects a safety-I perspective regarding patient safety, with focus on incidents that could have or did lead to harm for patients during healthcare, assuming that safety is achieved by eliminating what can go wrong.[3] This perspective assumes that adverse outcomes are caused by identifiable failures or malfunctions of specific components different from situations when things go right.[3 4] Similarly, root cause analysis (RCA) has become one of the most widespread tools used in the investigation of healthcare-related incidents, and presumes that such incidents can be explained by linear cause–effect chains.[5 6] Determining what had happened and why an incident occurred should not be the final goal of an incident investigation; the identification of gaps in

service provision and means of improving relevant areas of the healthcare organisation are important for improving safety.[7] To successfully learn from past incidents, methods to sustainably record and share relevant data are essential.[8 9] However, prior studies have shown that, in healthcare, post-incident investigations usually provide little learning beyond the staff and units involved.[10 11] Thus, the actual value of incident-reporting systems and the RCA approach in healthcare has been questioned.[8 12–15] With the introduction of new concepts in patient safety, such as safety-II and resilient healthcare, new approaches for improving healthcare have focused on learning from all occurrences in daily practice: to identify both those factors that support a good outcome and those that increase the risk of patient harm.[3 4] In the concept of safety-II, focus is on 'work in practice', that is, to better understand how clinicians provide good quality healthcare in real-time dynamic systems, including the interactions between patient care, environmental contexts and healthcare culture. In this perspective, safety is achieved through understanding healthcare staff's adaptations to varying conditions and ensuring that as much as possible goes well.

Swedish law states that events with severe patient harm, as well as events involving risk of severe patient harm, that could have been avoided if appropriate actions had been taken by healthcare professionals, should be reported to the supervisory authority.[16] This report to the authority should be preceded by an investigation, conducted by the healthcare providing organisation, of the healthcare services provided to the patient before the adverse event. The content of the investigation is regulated by law, and requires identification of the contributory causes of the incident and of service improvements that may prevent the reoccurrence of such an incident.

Suicide is a global health problem with an estimated 800 000 deaths worldwide every year.[17] Suicidal behaviours are heterogeneous and complex and influenced by several interacting biological, genetic, psychological, social, environmental and situational factors over time.[18] A large proportion of the individuals who die from suicide have contact with healthcare professionals close in time before their deaths.[19 20] Post-suicide studies have found that the vast majority of suicide victims have psychiatric illnesses at the time of their deaths.[21–23] This suggests that healthcare professionals play an important role in suicide prevention.[24] However, the nature of suicide as a process going on over time, usually occurring outside the hospitals without any witnesses nor staff around, make suicide as a case of patient harm, somewhat different from most other kinds of such incidents in healthcare. Few studies have applied patient safety paradigms to advance understanding of preventing suicide[25] although there are examples of studies of health services associated with reductions in suicide rates, such as well-developed community outpatient services[26] and the implementation of 24-hour crisis services.[27] Kapur et al suggest system-wide changes implemented across the patient care pathway

could be a key strategy for improving patient safety in mental healthcare.[28]

In an effort to understand whether failures in any area of the healthcare system have contributed to suicide, and in an attempt to improve suicide-prevention, the Swedish National Board of Health and Welfare in 2006 stipulated that all suicides that occur among patients who were receiving healthcare or were in contact with healthcare services within the 4 weeks preceding the event must be reported to the authority by the healthcare provider.[29] This remained mandatory regardless of whether the provider determined the suicide to be preventable. In September 2017, this regulation was updated to state that only suicides regarded as 'severe patient harm' (ie, preventable) must be reported to the supervisory authority.[30]

Before 2011, the supervisory authority performed their own investigations of incidents, and had the power to reprimand the provider and responsible staff. The role of the supervisory authority changed in 2011, when the Swedish Patient Safety Act (2010:659)[16] was implemented. This new law made healthcare organisations responsible for patient-safety improvement, and the role of the supervisory authority was changed to review the investigations made by the providers, and ensure that they were satisfactorily fulfilled and that appropriate actions had been taken to ensure a high level of patient safety. In particular, the authority determines whether the healthcare provider has fulfilled their legislated duties, or whether there are shortcomings in the investigation, in which case the authority may recommend revisions or conduct a site visit to inspect the healthcare provider.

To our knowledge, there are no published evaluations of the outcomes of investigations of specific types of patient harm over time, here exemplified by suicide.

The objective of this study was to explore how mandatory reporting of suicide cases as incidents of potential patient harm has influenced the investigations of healthcare systems. To perform this, a 13-year perspective was adopted, and the lessons and possible improvements for patient safety regarding suicide prevention were examined.

## METHODS

This study followed the guidelines of the STROBE (Strengthening the Reporting of Observational Studies in Epidemiology) checklist for reporting observational studies, available as an online supplemental file 1.

### Cases

Three cohorts of suicide cases, each from a different time period, that were reported to the supervisory authority were chosen for analysis. Cohort 1 comprised the cases reported to the supervisory authority in 2006, from the time the reporting of suicides became mandatory, to 2007 (n=279). Cohort 2 comprised all suicides reported in 2015, this represented a period when mandatory reporting was well established among healthcare providers (n=436). Cohort 3 comprised all reported suicides from

1 September 2017, which was the time the law regarding reporting was changed, to 30 November 2019 (n=316).

Complete reports of the incident investigations conducted by the healthcare providers with associated patient records and the subsequent decisions of the supervisory authority were obtained from the supervisory authority, granted by a contract of secrecy. Every individual suicide case was given a code number and the patient's demographic data and treatment received in the months preceding his/her death were registered. Major diagnoses were documented and coded in accordance with the International Statistical Classification of Diseases and related Health Problems, 10th revision (ie, ICD-10).

## Categorisation of data

A coding scheme was used to categorise the contributory causes of the respective suicides, the actions reported in the investigations and the decisions of the authority. The same coding scheme was used in a prior study of reported suicide cases in Sweden.[11] This scheme is based on the general categories used in the most common method of investigating adverse events in Swedish healthcare, which is in turn based on RCA.[31] To make the categorisation more specific, four of the major categories were divided into additional subcategories. Every category was described and exemplified and a category of 'others' was added in case none of the other categories was considered appropriate. In this present study, the contributory causes were reported as 'deficiencies'. Meanwhile, an 'action' was defined as any intervention performed in attempt to prevent new suicides: therefor, actions taken to prevent reported suicides (telephone calls, resuscitations) or actions aimed at informing family members or staff that a suicide had occurred were not registered as actions in this study. Separate notes were made when a deficiency or action was related to a healthcare-service routine, as well as in regard to how learning from the investigation was described. To ensure consistency, all data collection and categorisation were conducted by only one researcher (EF), a psychiatrist with extensive experience in patient-safety issues.

## Organisational levels

Classification of the organisational levels of deficiencies and actions was conducted to better understand where in the organisational system the identified deficiencies and actions were situated. The deficiencies and actions were coded based on a micro–meso–macro perspective.[32] Microsystems were defined as the basic elements of the healthcare services provided for the patient, such as the inpatient or outpatient care unit. The mesosystem encompassed interactions between different microsystem units, such as cooperation between departments or different healthcare providers. The macrosystem involved the entire healthcare system, such as legislation, political prioritisations and national policies on healthcare. For each case, the highest organisational level for each deficiency and action was coded.

## Supervisory authority

The mandate stipulated to the authority by legislation differed between cohort 1 and cohorts 2 and 3, hence the formulation of the decisions also differed. In this paper, to facilitate comparison among these outcomes, for all cohorts only decisions categorised as 'immediate approval' and 'inspection' were noted, as these remained unchanged. A note was made if a physician employed by the supervisory authority was involved in the decision-making.

## Statistical analyses

Frequencies for each category, organisational hierarchal level of deficiencies and actions and decisions of the supervisory authority were analysed per individual and aggregated per cohort.

$X^2$ tests of independence were used to compare the number of new routines and the absence of routines within the same cohort, as well as the proportion of the organisational hierarchy of deficiencies and actions between cohorts. We considered a two-sided p value of <0.05 to indicate statistical significance. As the prerequisites differed between the cohorts, no further statistical analyses to compare the cohorts were judged to be possible.

The statistical analyses were performed using IBM SPSS Statistics V.24.

## Patient and public involvement

Patients or public were not involved in this study.

## RESULTS

### Cases

Demographic data for the cases showed similarities across the cohorts, with a dominance of men and a majority of cases reported by psychiatric care. One-fourth of the cases died from suicide within 1 day of their last contact with a healthcare professional; half of the cases died from suicide within 2–4 days of their last contact. For details, see table 1.

### Deficiencies in healthcare

Cohort 3 showed the largest proportion of cases for which deficiencies in healthcare were considered to have contributed to the suicide. In this cohort, only suicide cases considered to involve severe patient harm could have been prevented if different actions had been taken by healthcare professionals were to be reported. Over time, some changes in the proportions for the categories of deficiencies were observed, but they remained centred on final patient contact with healthcare services. In cohorts 1 and 2, the most common deficiencies concerned 'suicide risk assessment'. In general, in cohort 1 these deficiencies related to an absence of local guidelines for suicide risk assessment, and in cohort 2 to nonadherence to existing guidelines. In cohort 3, deficiencies in 'treatment' and 'external communication' were the

**Table 1** Characteristics of the suicide cases reported to the supervisory authority across the three cohorts

| | | Cohort 1 (n=279) | Cohort 2 (n=436) | Cohort 3 (n=316) |
|---|---|---|---|---|
| **Characteristic** | | | | |
| Age, years | Range | 15–95 | 13–93 | 11–95 |
| | Percentile 25 | 36 | 33 | 29 |
| | Percentile 50 | 50 | 49 | 42 |
| | Percentile 75 | 64 | 61 | 57 |
| Gender | Men | 166 (60) | 283 (65) | 213 (67) |
| | Women | 113 (40) | 152 (35) | 103 (33) |
| Reporting healthcare service | Psychiatric care | 195 (70) | 290 (67) | 233 (74) |
| | Primary care | 47 (17) | 94 (22) | 56 (18) |
| | Somatic care | 21 (7) | 33 (8) | 16 (5) |
| | Other | 16 (6) | 18 (4) | 11 (3) |
| Days between last contact with healthcare services and death | Range | 0–70 | 0–88 | 0–240 |
| | Percentile 25 | 0 | 1 | 0 |
| | Percentile 50 | 2 | 4 | 3 |
| | Percentile 75 | 7 | 10 | 9 |
| Receiving inpatient care at time of death | | 45 (16) | 36 (8) | 44 (14) |
| Receiving compulsory psychiatric treatment at the time of death* | | 15 (5) | 22 (5) | 20 (6) |
| Major psychiatric diagnosis documented and coded in accordance with ICD-10 in patients' records | Total (F00–F98) | 228 (82) | 371 (85) | 288 (91) |
| | Affective disorder (F30) | 119 (43) | 153 (35) | 105 (33) |
| | Anxiety disorder (F40) | 35 (13) | 77 (18) | 60 (19) |
| | Substance abuse (F10) | 29 (10) | 51 (12) | 37 (12) |
| | Psychosis (F20) | 22 (8) | 36 (8) | 30 (10) |
| | Personality disorder (F60) | 12 (4) | 13 (3) | 13 (4) |
| | Attention deficit disorder (F90) | 1 (0) | 13 (3) | 12 (4) |
| | Autism spectrum (F84) | 3 (1) | 13 (3) | 9 (3) |
| | Other | 7 (2) | 15 (3) | 22 (7) |
| Suicide-risk assessment documented in patients' records in the 3 months before death | Absent | 135 (49) | 108 (25) | 119 (38) |
| | Low | 61 (22) | 171 (39) | 91 (29) |
| | Elevated, not acute | 61 (22) | 116 (27) | 75 (24) |
| | High/acute | 19 (7) | 41 (9) | 31 (10) |
| Prior suicide attempt | | 120 (46) | 204 (47) | 154 (49) |
| Suicide method | Hanging | 112 (40) | 160 (37) | 128 (41) |
| | Intoxication | 42 (15) | 110 (25) | 53 (17) |
| | Jumping | 21 (8) | 13 (3) | 19 (6) |
| | Train | 11 (4) | 35 (8) | 22 (7) |
| | Drowning | 15 (5) | 28 (6) | 13 (4) |

Continued

**Table 1** Continued

| | | Cohort 1 (n=279) | Cohort 2 (n=436) | Cohort 3 (n=316) |
|---|---|---|---|---|
| | Shooting | 10 (4) | 27 (6) | 14 (4) |
| | Others | 13 (8) | 12 (3) | 16 (5) |
| | Not reported | 51 (18) | 50 (12) | 51 (16) |
| Location of suicide | Home | 154 (56) | 248 (57) | 161 (51) |
| | Hospital | 23 (8) | 22 (5) | 33 (10) |
| | Other | 53 (19) | 131 (30) | 83 (26) |
| | Not reported | 44 (16) | 35 (8) | 39 (12) |

The data in the table comprise numbers and percentages, n (%).
Cohort 1: cases reported in 2006–2007, cohort 2: cases reported in 2015 and cohort 3: cases reported in 2017–2019.
*Includes both inpatient and outpatient compulsory treatment.
ICD-10, International Classification of Diseases and related Health Problems, 10th revision.

most common. Examples of deficiencies in 'treatment' were delayed, or a lack of, follow-up after prescription of medication, or non-adherence to treatment guidelines. Examples of deficiencies in 'external information' were a lack of or insufficient information exchange between healthcare providers. For details, see table 2.

### Proposed actions for addressing deficiencies

In a majority of the cases, the providers proposed actions for improving the healthcare services. The proportions of the action categories differed between the cohorts. In cohort 1, actions relating to 'suicide risk assessment' were most common, usually involving the creation of new local guidelines regarding this issue. In cohorts 2 and 3, actions centred on education, present in more than half of the cases. Examples of educational actions were reminding staff about existing local guidelines, holding case-report discussions at staff meetings and staging lectures regarding suicide risk assessment. For details, see table 3.

### Learning and sharing

Any lessons learnt and the sharing of experiences obtained from cases and investigations usually remained within the department in question. Sharing outside the department was reported in 4% (n=17) of the cases in cohort 2, and in 7% (n=21) of the cases in cohort 3. Sharing outside the department was not reported in any cases in cohort 1.

### Routines

Over time, proposals for actions concerning updating or developing new routines became more common in the investigations. In cohorts 2 and 3, there were significantly more cases featuring the proposed development of new routines when compared with the number of cases for which an absence of routines was identified. In all cohorts, the number of revisions exceeded the number of identified dysfunctional routines. Non-adherence to existing routines was highlighted in almost one-third of the cases in cohort 3. For details, see table 4.

**Table 2** Proportions of cases with deficiencies, as reported in the post-suicide investigations of the healthcare services' actions

| | Cohort 1 (n=279) | Cohort 2 (n=436) | Cohort 3 (n=316) |
|---|---|---|---|
| Cases with deficiencies, total | 136 (49) | 240 (55) | 248 (78) |
| Category | | | |
| Communication and information | | | |
| Communication with peers and family members | 8 (3) | 51 (12) | 39 (12) |
| Documentation | 57 (20) | 65 (15) | 68 (22) |
| External communication | 21 (8) | 74 (17) | 91 (29) |
| Internal communication | 18 (7) | 61 (14) | 68 (22) |
| Education and competence | | | |
| Education and competence not specified | 12 (4) | 54 (11) | 50 (16) |
| Education and competence in suicide risk assessment | 5 (2) | 9 (2) | 13 (4) |
| Organisation and management | | | |
| Human resources | 15 (5) | 60 (14) | 53 (17) |
| Number of beds | 2 (1) | 9 (2) | 5 (2) |
| Organisation/management | 2 (1) | 13 (3) | 13 (4) |
| Policies and procedures | | | |
| Treatment | 26 (9) | 84 (19) | 92 (29) |
| Suicide risk assessment | 92 (33) | 86 (20) | 76 (24) |
| Work process | 20 (7) | 50 (11) | 51 (16) |
| Diagnostics | 16 (6) | 54 (12) | 41 (13) |
| Care plan and crisis plan | 10 (4) | 46 (11) | 53 (17) |
| Technics and equipment | 5 (2) | 13 (3) | 15 (5) |
| Other | 2 (1) | 11 (3) | 0 (0) |

The data in the table comprise numbers and percentages, n (%).
Cohort 1: cases reported in 2006–2007, cohort 2: cases reported in 2015 and cohort 3: cases reported in 2017–2019.

## Organisational hierarchy

For both deficiencies and proposed actions, the microsystem perspective remained dominant over the 13-year period. However, cohorts 2 and 3 showed a significant increase in the proportion of deficiencies and actions at the mesosystem level compared with cohort 1. No deficiencies were found at the macrosystem level. For details, see table 5.

Examples of deficiencies at the microsystem level were inadequacies in doctors' prescriptions or in suicide-risk assessments. Examples of actions at the microsystem level were case discussions at staff meetings, lectures and the development of new checklists. Deficiencies at the mesosystem level included shortcomings in cooperation between the psychiatric clinic and somatic clinic, or inadequate communication between the hospital and primary care centre. Examples of actions at the mesosystem level were alterations of procedures for communication or cooperation between different healthcare providers.

**Table 3** Proportions of cases for which actions were recommended in the post-suicide investigations

| | Cohort 1 (n=279) | Cohort 2 (n=436) | Cohort 3 (n=316) |
|---|---|---|---|
| Cases with actions, total | 133 (48) | 346 (79) | 283 (90) |
| Category | | | |
| Communication and information | | | |
| Communication with peers and family | 12 (4) | 51 (12) | 27 (9) |
| Documentation | 39 (14) | 71 (16) | 65 (21) |
| External communication | 22 (8) | 80 (18) | 83 (26) |
| Internal communication | 15 (5) | 55 (13) | 46 (15) |
| Education and competence | | | |
| Education and competence not specified | 35 (13) | 166 (38) | 136 (43) |
| Education and competence in suicide risk assessment | 44 (16) | 136 (31) | 85 (27) |
| Organisation and management | | | |
| Human resources | 7 (3) | 67 (15) | 42 (13) |
| Number of beds | 1 (0) | 4 (1) | 1 (0) |
| Organisation/management | 6 (2) | 22 (5) | 20 (6) |
| Policies and procedures | | | |
| Treatment | 21 (8) | 56 (13) | 64 (20) |
| Suicide risk assessment | 74 (27) | 94 (22) | 51 (16) |
| Work process | 28 (10) | 119 (27) | 87 (28) |
| Diagnostics | 8 (3) | 28 (6) | 25 (8) |
| Care plan and crisis plan | 6 (2) | 46 (11) | 51 (16) |
| Technics and equipment | 12 (4) | 22 (5) | 22 (7) |
| Other | 1 (0) | 8 (2) | 3 (1) |

The data in the table comprise numbers and percentages, n (%).
Cohort 1 comprises of cases reported in 2006–2007, cohort 2 cases reported in 2015 and cohort 3 cases reported in 2017–2019.

## Decisions of the supervisory authority

In all cohorts, the majority of the reports from the healthcare providers were approved by the supervisory authority without further requirements. Immediate approval was provided for 59% (n=164) of the reports for cohort 1,

**Table 4** Deficiencies and actions in routines, reported in the post-suicide investigations

| | | Cohort 1 (n=279) | Cohort 2 (n=436) | Cohort 3 (n=316) |
|---|---|---|---|---|
| Routines, deficiencies | Non-adherence | 10 (4) | 44 (10) | 95 (30) |
| | Absent | 38 (14) | 30 (7) | 28 (9) |
| | Dysfunctional | 1 (0) | 0 (0) | 8 (3) |
| Routines, actions | Revision | 24 (9) | 58 (13) | 47 (15) |
| | New | 55 (20) | 94 (22)* | 99 (31)* |

The data in the table comprise numbers and percentages, n (%).
Cohort 1: cases reported in 2006–2007, cohort 2: cases reported in 2015 and cohort 3: cases reported in 2017–2019.
*Significantly more cases involved the development of new routines when compared with the number of absent routines, p<0.001.

65% (n=284) for cohort 2 and 59% (n=186) for cohort 3. Meanwhile, inspections of the healthcare provider occurred for 9% (n=25) of the cases in cohort 1, 6% (n=25) of those in cohort 2% and 4% (n=13) of those in cohort 3. A physician employed at the supervisory authority was involved in the decision-making for 89% (n=249) of the cases in cohort 1, in 4% (n=17) of the cases in cohort 2 and 13% (n=40) of the cases in cohort 3.

## DISCUSSION

This study explored changes in the outcomes of post-suicide investigations by healthcare services in cases reported as potential incidents of patient harm, adopting a 13-year perspective. Possible improvements for patient safety that could contribute to suicide prevention were also examined in the context of these reports.

Over time the investigations generally and consistently focused on final patient contact, analysing the immediate interface between the patient and staff from a microsystem level perspective.

The most common measures recommended for all cohorts were updating existing or developing new routines, and educational actions—potentially unsustainable, person-based. Sharing conclusions across departments was planned in only a small percentage of the cases. This similarity of investigation outcomes over the years, regardless of changes in legislation, suggests that the investigations were adapted to suit the structure of the authority report rather than specific incidents, and imply that no new service improvements or lessons are being identified.

The suicide rate in Sweden has not shown any obvious decline since the reporting of all suicide cases became mandatory,[33] and the healthcare-service deficiencies highlighted in these reports as being of significance continue to occur. In other words, despite several thousand investigations into healthcare performance prior to suicides over the last few decades, aimed at identifying

actions to improve healthcare for patients with suicidal tendencies, the same contributing factors remain.[34] This suggests that the actions taken to date have not been sufficient. A possible means of addressing this would be the systematic aggregation and analysis of trends through multiple investigations, which would help to conclusively identify recurrent deficiencies, and encouraging investigators to act as facilitators of organisational development instead of mandating single investigations.[35] Another explanation could be that the current investigations fail to identify significant deficiencies, suggesting we need to develop more sophisticated methods for investigations of suicide.

Most of the reported cases in this study had their last contact with a healthcare professional within days of their deaths. Data in this study represent a subset of the total deaths by suicide, excluding these not reported to the authority. However, during the last 3 years of mandatory reporting (2014–2016), 51%–58% of the total suicides in Sweden were reported per year to the supervisory authority.[33 34] Two-thirds of the cases lacked a documented report of an elevated risk of suicide in the months before the death, and this persisted across cohorts, despite the strong focus in many of the analysed investigations on actions related to suicide-risk assessment and education in this issue. Over the years, there has been a shift from reports of an absence of local policies for suicide-risk assessment to reports of non-adherence to existing policies for suicide-risk assessment. In the studied cohorts, only 7%–10% of the patients were documented as being at high risk of suicide during the last months before death. Studies have shown that suicide risk instruments and risk scales do not enable clinicians to predict which patients will die by suicide,[36–38] raising the question of the value of these assessments.[39] In an interview study healthcare professionals describe they set forms and checklist aside to prioritise trust during suicide risk assessment.[40]

Approximately half of the suicide victims in all cohorts had a documented prior suicide attempt, and it is shown that previous suicide attempt, especially repeated, imply higher risk for suicide persisting over decades.[41] Learning from cases of the successful treatment of patients who have survived prior suicidal crises could thus be of importance for improving suicide prevention in healthcare. However, such learning actions are not recommended in the Swedish reporting system, which is currently based on a safety-I model; thus possible learning opportunities are not supported unless a safety-II perspective is supplemented.[3]

Cohort 3 showed a higher proportion of deficiencies in 'education and competence' when compared with cohorts 1 and 2. These deficiencies were often connected to deficiencies in 'human resources' and 'internal communication', suggesting difficulties in recruiting personnel with adequate competence, shortcomings in the introduction of new staff and complications integrating locum doctors.

Deficiencies in 'external communication' and 'treatment' were present in almost one-third of the cases

**Table 5** Respective distributions of the highest organisational hierarchy levels for the deficiencies and actions associated with the cases

| | | Cohort 1 | Cohort 2 | Cohort 3 |
|---|---|---|---|---|
| Organisational level, deficiencies | Micro | 121 (90) | 157 (65) | 179 (73) |
| | Meso | 13 (10) | 83 (35)* | 67 (27)* |
| | Macro | 0 (0) | 0 (0) | 0 (0) |
| Organisational level, actions | Micro | 115 (85) | 225 (65) | 206 (75) |
| | Meso | 20 (15) | 120 (35)* | 70 (25)* |
| | Macro | 0 (0) | 1 (0) | 0 (0) |

Only the highest level for each case is noted. The data in the table comprise numbers and percentages, n (%). Cohort 1: cases reported in 2006–2007, cohort 2: cases reported in 2015 and cohort 3: cases reported in 2017–2019.
*Significantly larger proportion of cases with deficiencies or actions at the mesosystem level when compared with cohort 1, p<0.005.

in cohort 3. This cohort showed a younger population with some higher degree of psychiatric diagnoses, which suggests that this was a more complex group with a need for support from different care providers, requiring external collaboration and, possibly, more complex treatment interventions.

In all cohorts, there was a pronounced focus on routines. Updating existing or developing new routines was the most common recommendation proposed in the investigations over the years. All cohorts, but most obviously cohort 3, showed a mismatch between the number of cases where an absence of routines was noted and the number of cases for which the development of new routines was recommended. Further, the number of revisions exceeded the number of identified dysfunctional routines. Non-adherence to existing routines was highlighted in almost one-third of the cases in cohort 3, and the solutions seemed to focus on creating new routines instead of ensuring adherence, preconditions and usability. Notably, reflections on why adherence to existing routines failed from a system perspective were missing in the investigations. This obsession with routines reflects the current predominant perspectives of safety-I. In the perspective of safety-II, the variability of performance conditions that is the reality in healthcare, requires that how the work is performed has to be adopted to the current specific situation to maintain safety.[3 4] Thereby, no precise detailed descriptions of how all work should be done in all situations is possible or even desirable.

Further, changes or re-implementation of routines are person-based and have weak efficacy from a systemic perspective, but require less effort than strong actions on a systemic level.[42 43] The same concerns were present regarding educational actions, which were highlighted in over half of the cases in cohorts 2 and 3. The dominance of person-based actions at the microsystem level is not unique for the Swedish setting. Kellogg *et al* obtained the same findings in a review conducted in the USA,[12] and other studies have reported that investigators complete their analyses after identifying human error, rather than proceeding to identify system-based problems.[44 45] Attributing issues to human error easily leads to person-based solutions, and creates a focus on what is possible rather than what is needed.[35] Recurrent widespread microsystem issues require whole-system responses at macro level to be solved.

Suicide locations and methods were similar in all cohorts, but were reported in less than 90% of the investigations in cohort 3. This was surprising, as these cases were regarded as representing incidents of severe patient harm, and analysis of the specific circumstances concerning the suicide should be of importance in regard to evaluating preventable factors.

The distribution of the supervisory authority's decisions remained similar over the years; most reports were approved without further arrangements. In a small number of cases, the authority made a site visit, but the frequency of such visits declined as time passed. Supervision can be a strong tool and incitement for improvement

and development of healthcare services,[14] but the results in this study suggest that the authority did not avail of this. Mandatory reporting thus was determined to be a process of information transfer between healthcare providers and the authority, rather than a means of creating a participative improvement that enhances safety for patients with suicidal tendencies.

The overall aim of the incident-reporting system is to make healthcare safer, which presupposes learning. However, learning that extends beyond the staff involved in the incident requires information sharing. The review of the reports in this study showed that sharing information between departments was planned in a low percentage of cases, which is in concordance with similar results reported in a previous Swedish study.[10] Learning is a complex social and participative process that involves people actively reflecting on and organising shared knowledge and practices.[8] Safety begins, rather than ends, with incident reports, and requires broad, in-depth and high-quality investigations and careful planning and follow-up of the implementation of corrective actions to ensure they are sustainable over time.[46] To generate persistent knowledge and learning from cases, feedback should include more than a passive, brief report in a staff meeting that reminds of or notifies of the updating of a routine.

Suicide is usually the final outcome of several interacting factors over time, and only a small proportion of suicides are committed in hospitals.[47 48] Most suicides occur in the patient's home without any witnesses or staff; this makes suicide, as a case of patient harm, somewhat different from most other kinds of such incidents in healthcare. The requirements of the report to the authority are the same for all kinds of incidents, meaning the investigating process may be adapted to suit the standard template rather than the specific character of the incident. Analysing the last contact with a healthcare professional from a microsystem level perspective is not sufficient to learn how healthcare can better help patients with suicidal tendencies. The investigation should integrate analysis of the suicidal process over time, including suicide prevention tools. To advance this issue, a shift in investigations requirements and reports is needed, as well as more sophisticated infrastructures for investigation, learning and sharing in healthcare services. Innovation based on relevant patient safety paradigms combined with suicide preventions research is needed.

### Limitations and strengths

All data were based on the healthcare providers' investigations and reports to the supervisory authority, a subset of the total deaths by suicide, excluding these not reported to the authority.

The content in the reports is regulated by law; however, the quality of analysis differs and there still may have been additional shortcomings and inadequacies that were not mentioned in the reports or observed by the authority, as well as there were actions mentioned which had no relevance in the circumstances described. Furthermore,

there is no national taxonomy for the categorisation of deficiencies and actions; a coding scheme created by the authors and used in a prior study was used. The category of 'other' was used only in a few cases, suggesting that the categories in the coding scheme covered most of the reported deficiencies and actions.

The strengths of this study are that all investigations concerned the same kind of incident; suicides, and the data were population-based. Further, all data collection and categorisation were conducted by only one researcher, who is a psychiatrist with experience working with patient safety issues; this made the categorisation vulnerable to bias, but ensured a high level of consistency.

## Conclusions

The mandatory reporting of suicides as potential cases of patient harm was shown to be restricted to information transfer between healthcare providers and the supervisory authority, rather than fostering participative improvement of patient safety for suicidal patients.

The similarity in outcomes across the cohorts, regardless of changes in legislation, suggests that the investigations were adapted to suit the structure of the authority's reports rather than the specific incident type, and that no new service improvements or lessons are being identified.

To develop more sophisticated infrastructures for investigation, learning and information sharing, it is necessary to learn more about preconditions and complexity in the analysis of suicides and the suicidal process.

A shift in investigations' recommendations and reports should be encouraged, to also include learning from successfully treated and resolved suicide-related crises.

**Acknowledgements**  The authors are grateful to Region Jönköpings county and Futurum for funding and to Public Health Agency of Sweden for support.

**Contributors**  EF designed the study, collected and registered the data, made the first analyses and wrote the manuscript. BAG, AR and ÅW contributed to the study design, analyses of the data and revisions of the manuscript. All authors read and approved the final manuscript.

**Funding**  This study was funded by Futurum, the research centre at Region Jönköping county.

**Competing interests**  None declared.

**Patient consent for publication**  Not required.

**Ethics approval**  According to the Swedish Act Concerning the Ethical Review of Research Involving Humans (2003:460) and an advisory opinion from the Regional Ethical Review Board (no. 2017/234), this study did not require an ethical review as it did not include human participants.

**Provenance and peer review**  Not commissioned; externally peer reviewed.

**Data availability statement**  Data are available upon reasonable request. Data may be obtained from a third party and are not publicly available. De-identified data may be available upon reasonable request from elin.froding@rjl.se. The data of the case reports were obtained from the supervisory authorities, granted by a contract of secrecy. The data can only be obtained by direct request to the supervisory authorities: for cohort 1: The national board of health and welfare, mail: Registerservice@socialstyrelsen.se, and for cohorts 2 and 3: Health and Social Care Inspectorate, mail: samordna.utlamnanden@ivo.se.

**ORCID iDs**
Elin Fröding http://orcid.org/0000-0002-9095-1322
Axel Ros http://orcid.org/0000-0001-6302-8068

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
