## [Reviewer comments · BMJ Open]

ARTICLE DETAILS

TITLE (PROVISIONAL)	Suicide as an incident of severe patient harm – a retrospective cohort study of investigations after suicide in Swedish healthcare in a 13-year perspective
AUTHORS	Fröding, Elin; Gäre, Boel Andersson; Westrin, Åsa; Ros, Axel

VERSION 1 – REVIEW

REVIEWER	Fredrik A. Walby National Center for Suicide Research and Prevention. Faculty of Medicine, University of Oslo, Norway
REVIEW RETURNED	18-Oct-2020

GENERAL COMMENTS	The objective of this study was to explore how mandatory reporting of suicide cases as incidents of potential patient harm has influenced the investigations of healthcare systems in Sweden. It is generally well written and addresses an important area that are seldom studied. The paper is relevant both from a psychiatric and a suicidologic as well as a patient safety perspective. The methods used are generally sound. I have some comment for improvement of the paper, particularly regarding the introduction and the discussion sections. Introduction: “Safety I-II”, and resilience are clearly relevant theoretical concepts here, but described in a very condensed matter. Please give some more description to help a reader not experienced in the recent patient safety literature to understand. The authors might also want to include some of the recent work by Siv Hilde Berg regarding suicide prevention in a safety-II perspective, see for example: https://pubmed.ncbi.nlm.nih.gov/32560682/. The introduction describes patient safety and the particular Swedish system in a clear and comprehensive way, but I miss a more general discussion of suicide and suicide prevention. Suicides are a patient safety issue of substantial importance in the Mental Health systems, but are there issues specific to suicide investigations as compared to other investigations of other unexpected events?. Further, are there relevant examples from other countries, or specific recommendations from the field of patient safety research pertinent to this issues? Discussion: Tbl1: Most of the suicides under investigation took place very close to the contact with health care. Please discuss if the material is representative of suicides connected to mental health care in general, and possible implications for the results of this study. I think the paper would be improved, and be of more relevance outside the Swedish context, if the authors could discuss their mostly sobering findings about the effect of the current system on
--

	patient safety in more detail and on a higher level: Is this way of trying to increase patient safety effective regarding suicide in health care? What could be alternatives or are there ways to improve the system for investigations? Are there other alternatives, such as the more data driven and group based work of the National Confidential Inquiry based in Manchester or other approaches? Although there is some discussion at the of such issues, more suggestions on how to proceed may be warranted. P 22 and following: Although suicide attempts and suicide share some risk factors, they are clearly different behaviors. It is not clear to me from the discussion how a greater weight on learning from attempts would improve suicide prevention in this population. Safety I-II, as mentioned in the introduction, should be discussed in more detail. Minor details: Line 7-8: been mandatory in Sweden since 2006-2017 ??? I guess this is explained on line 60, but should be rewritten to be understandable on its own in the abstract. P 18: low proportion of suicide under compulsory care indicate that this fulfills its purpose. That cannot be answered with this design and it's not relevant for the papers aim. I would suggest removing this speculation. The issue with limitations by using only one person to collect and categorize data are correctly stated as a limitation on line 8-9. This is not so clear from discussion of limitations at the end of the paper.
--	--

REVIEWER	Michael Smith NHS Greater Glasgow & Clyde, UK
REVIEW RETURNED	18-Dec-2020

GENERAL COMMENTS	P3 lines 19, 33 – It would be helpful to reference explain what Safety-I & II mean in this context. eg https://www.tandfonline.com/doi/full/10.1080/09638237.2020.1714013 P3 49 some would contest the interpretation that 90% of people had a psychiatric illness at the time of death, or that any illness was the main cause of death. Similarly, while healthcare professionals may try their best to prevent suicide, there is little evidence to show that this is effective. P8 – worth noting that there is little or no evidence that carrying out suicide risk assessments can reduce suicide – even though all systems focus on this. P10, 18 – It is striking that so little learning was shared outside the department – not clear whether this was because they were mainly “micro” issues, or whether this itself reflected a failure of communication. Similarly (line 52) the lack of “macro system” problems seems an unlikely outcome. From an organisational perspective, persistent and widespread “micro” issues (eg prescriptions, communication, risk assessments) should become macro issues since they require whole-system responses. Discussion P11, lines 53-56 – I wasn't clear about the wording about “similarity of investigations”... is it arguing that the investigations were primarily
---

about meeting organisational objectives (“ticking a box”) rather than seeking to fully understand events?

P12, line 3 – it’s interesting that there’s no correlation between suicide investigations and the suicide rate (though of course too many confounders to draw conclusions). But the problem might be either (as suggested) that the actions weren’t implemented adequately, or (at least as important in my view) that the investigations were structured in such a way as to fail to identify the true root causes.

P12, line 12 – other work shows that about 80% of suicides were considered “low risk” at the time of death.

P13, line 31 – suicide is different from other kinds of patient harm in many ways! In fact, the assumption that there are significant similarities may be a fundamental problem with these reviews.

The limitations set out on page 13 are accurate, but I think the authors could be more explicit about the broader context, eg:

- In the UK, about 1/3 of people who died by suicide had no contact with the healthcare system, and only about ¼ were in contact with MH services. Presumably similar proportions apply in Sweden, so this study is (quite reasonably) only able to examine a subset of the total deaths by suicide.
- The study only captures the perspective of the system (no external referencing to changes in suicide outcome, for example, or family and carer perspectives).
- This study therefore can’t identify either “false positives” – recommendations that were made without any relevance – or “false negatives” – items that were missed.
- Contact within 4 weeks and help-seeking only within healthcare settings limits the possible conclusions that can be drawn – it will necessarily only be concerned with “final acts and omissions” in a clinical context, rather than a much longer trajectory of care and life events. Note this is mentioned in p13, line 35.
- Investigations which take place in the context of seeking to identify “failures in any area of the healthcare system” (p3, line 52) may not be conducive to open reporting by that same system... I note this point was recognised on page 13, line s 13-15.
- Depending on one investigator to categorise cases will enhance consistency, but may have a limited or idiosyncratic perspective.
- “Actions” to reduce suicides such as risk assessment and education may not lead to reduction in suicides, and implementation is often incomplete or ineffective. The actions may also have a function in assuaging anxieties and risk *for the system* while they ostensibly also seek to reduce suicide per se.

The discussions should include reference to the work of the UK National Confidential Inquiry into Suicide and Homicide, which was able to draw conclusions about the effectiveness of suicide prevention actions, eg:

Kapur N, Ibrahim S, While D, Baird A, Rodway C, Hunt IM, et al. Mental health service changes, organisational factors, and patient suicide in England in 1997–2012: a before-and-after study. *Lancet Psychiatry* 2016;3: 526–34

	And also the impact of hospitalisation on risks: Kapur N, Steeg S, Turnbull P, Webb R, Bergen H, Hawton K, et al. Hospital management of suicidal behaviour and subsequent mortality: a prospective cohort study. Lancet Psychiatry 2015;2: 809–16. I think that organisational psychology and dynamics have a significant influence on what is measured, reported and changed (or not changed). Although beyond the scope of this paper, it would be useful to see that aspect of risk management acknowledged. I think this is a thorough and worthwhile paper and would support publication, subject to the observations above. The limitations acknowledged by the authors and which I've referenced above are not only for this paper, but for this field of enquiry as a whole. I'd therefore politely suggest that the paper would be enhanced by an accompanying editorial setting out the need to critically examine the purpose of these reports, and to make the case for extending the "frame" for this work to include a much broader set of influences.
--	--

VERSION 1 – AUTHOR RESPONSE

Thank you so much for your helpful and wise comments! We found them all relevant and reasonable, and made revisions in the paper in concordance with your suggestions and our comments below.

Introduction:

"Safety I-II", and resilience are clearly relevant theoretical concepts here, but described in a very condensed matter. Please give some more description to help a reader not experienced in the recent patient safety literature to understand. The authors might also want to include some of the recent work by Siv Hilde Berg regarding suicide prevention in a safety-II perspective, see for example: <https://pubmed.ncbi.nlm.nih.gov/32560682/>.

Thank you for these suggestions! We have added some more information about Safety-I&II in the background to make it more understandable (p 3: 14-15, 29-33). We have read the recommended paper and found it relevant as a reference in the discussion (p 11: 37-39 ref 49).

The introduction describes patient safety and the particular Swedish system in a clear and comprehensive way, but I miss a more general discussion of suicide and suicide prevention. Suicides are a patient safety issue of substantial importance in the Mental Health systems, but are there issues specific to suicide investigations as compared to other investigations of other unexpected events? Further, are there relevant examples from other countries, or specific recommendations from the field of patient safety research pertinent to this issues?

Thank you for these relevant points. We have added this to background and some new references (p 3: 42-44, 48-50, p 4: 1-6, ref 18, 24-28).

Discussion:

Tbl1: Most of the suicides under investigation took place very close to the contact with health care. Please discuss if the material is representative of suicides connected to mental health care in general, and possible implications for the results of this study.

From available statistics from the supervisory authority 2014-2016 in Sweden, during the period of mandatory reporting to supervisory authority of cases in contact with healthcare in four weeks before death, we find that 51-58% of all suicides in Sweden were reported. We complemented our paper as suggested, in discussion and limitations (p 11: 27-29, p 13: 25).

I think the paper would be improved, and be of more relevance outside the Swedish context, if the authors could discuss their mostly sobering findings about the effect of the current system on patient safety in more detail and on a higher level: Is this way of trying to increase patient safety effective

regarding suicide in health care? What could be alternatives or are there ways to improve the system for investigations? Are there other alternatives, such as the more data driven and group based work of the National Confidential Inquiry based in Manchester or other approaches? Although there is some discussion at the of such issues, more suggestions on how to proceed may be warranted. *Thank you for these suggestions. The excellent work of National Confidential Inquiry is absolutely relevant here. As we conclude in the paper, we cannot see how we can come any further from here with the current methods of investigations, since no new service improvements or lessons are being identified. We suggest we should develop new methods for investigations and also use them in a different way. A systematic aggregation and analysis of trends through multiple investigations, would help to conclusively identify recurrent deficiencies, and encourage investigators to act as facilitators of organizational development and implementation of evidence based improvement in healthcare. We have made some adjustments in the paper in line with this (p 11: 22-24, p 12: 32, p 13: 20-21).*

P 22 and following: Although suicide attempts and suicide share some risk factors, they are clearly different behaviors. It is not clear to me from the discussion how a greater weight on learning from attempts would improve suicide prevention in this population.

Of course this can be discussed. Our suggestion is based on the fact that at least 40% of suicide victims have a history of prior attempt, being the most important risk factor for completed suicide and it is shown that previous suicide attempt, especially repeated, imply higher risk for suicide persisting over decades. Learning from prior close calls, is in line with thoughts of safety-II. We complemented and added a reference about this (p 11: 44-46, ref 41).

Safety I-II, as mentioned in the introduction, should be discussed in more detail.

Fine, we have added some reflections of these perspectives in the discussion (p 12: 19-23).

Minor details:

Line 7-8: been mandatory in Sweden since 2006-2017 ??? I guess this is explained on line 60, but should be rewritten to be understandable on its own in the abstract.

Thank you, we tried to make this more understandable (p 2: 3-4).

P 18: low proportion of suicide under compulsory care indicate that this fulfills its purpose. That cannot be answered with this design and it's not relevant for the papers aim. I would suggest removing this speculation.

Thank you for this notation! We removed this sentence (p 11: 41-43).

The issue with limitations by using only one person to collect and categorize data are correctly stated as a limitation on line 8-9. This is not so clear from discussion of limitations at the end of the paper.

Correct, thank you, we have added this to the limitations at the end of the paper (p 13: 35-36).

Reviewer 2

P3 lines 19, 33 – It would be helpful to reference explain what Safety-I & II mean in this context. eg <https://www.tandfonline.com/doi/full/10.1080/09638237.2020.1714013>

Thank you for this excellent article! We read it with great pleasure and added some more descriptions of S-I & II in the introduction (p 3: 14-15, 29-33).

P3 49 some would contest the interpretation that 90% of people had a psychiatric illness at the time of death, or that any illness was the main cause of death. Similarly, while healthcare professionals may try their best to prevent suicide, there is little evidence to show that this is effective.

Yes, you are right, we have revised the interpretation and added references (p 3: 46, ref 22-23).

P8 – worth noting that there is little or no evidence that carrying out suicide risk assessments can reduce suicide – even though all systems focus on this.

Correct and relevant, thank you. We added this with some new references to the discussion (p 11:36-39, ref 36-40).

P10, 18 – It is striking that so little learning was shared outside the department – not clear whether this was because they were mainly “micro” issues, or whether this itself reflected a failure of communication.

We agree with you. Similar results were reported in a previous Swedish study. We added this to the discussion (p 13: 2-3).

Similarly (line 52) the lack of “macro system” problems seems an unlikely outcome. From an organisational perspective, persistent and widespread “micro” issues (eg prescriptions, communication, risk assessments) should *become* macro issues since they require whole-system responses.

We really agree with you, this connection is usually missing, in line with the conclusions. We added this to the Discussion (p 12: 32)

Discussion

P11, lines 53-56 – I wasn’t clear about the wording about “similarity of investigations”... is it arguing that the investigations were primarily about meeting organisational objectives (“ticking a box”) rather than seeking to fully understand events?

Yes, we found that the investigations were adapted to suit the structure of the authority’s reports rather than the specific incident type.

P12, line 3 – it’s interesting that there’s no correlation between suicide investigations and the suicide rate (though of course too many confounders to draw conclusions). But the problem might be either (as suggested) that the actions weren’t implemented adequately, or (at least as important in my view) that the investigations were structured in such a way as to fail to identify the true root causes.

We agree with you, and made some clarifications and additions to this part (p 11: 22-24).

P12, line 12 – other work shows that about 80% of suicides were considered “low risk” at the time of death.

This highlights the problems with suicide risk assessment... we have added a sentence in this issue and some references (p 11: 36-37, ref 36-38).

P13, line 31 – suicide is different from other kinds of patient harm in many ways! In fact, the assumption that there are significant similarities may be a fundamental problem with these reviews.

We agree with you, and added a few sentences on this issue in the background as well (p 3: 48-50, p 4 1-6).

The limitations set out on page 13 are accurate, but I think the authors could be more explicit about the broader context, eg:

□ In the UK, about 1/3 of people who died by suicide had no contact with the healthcare system, and only about ¼ were in contact with MH services. Presumably similar proportions apply in Sweden, so this study is (quite reasonably) only able to examine a subset of the total deaths by suicide.

Right, we added this clarification to both discussion and limitations (11: 27-29, p 13: 25).

□ The study only captures the perspective of the system (no external referencing to changes in suicide outcome, for example, or family and carer perspectives).

Yes, these were the conditions for this study, as described in limitations (p 13: 24-28).

□ This study therefore can’t identify either “false positives” – recommendations that were made without any relevance – or “false negatives” – items that were missed.

Yes, these were the conditions for this study, and we made a clarification (p 13: 28-29).

□ Contact within 4 weeks and help-seeking only within healthcare settings limits the possible conclusions that can be drawn – it will necessarily only be concerned with “final acts and omissions” in a clinical context, rather than a much longer trajectory of care and life events.

Note this is mentioned in p13, line 35.

This is an important point, and one of the reasons that we think the concept of investigations of suicide has to be adjusted and adapted to the complexity of the suicidal process.

□ Investigations which take place in the context of seeking to identify “failures in any area of the healthcare system” (p3, line 52) may not be conducive to open reporting by that same system... I note this point was recognised on page 13, line s 13-15.

We agree with you.

□ Depending on one investigator to categorise cases will enhance consistency, but may have a limited or idiosyncratic perspective.

Yes, you are right. We have discussed this in our research team, and also made a minor double-blind test to check we categorized in concordance, which we did. We added a comment on that in Limitations (p 3: 4-5, p 13: 35-36).

□ “Actions” to reduce suicides such as risk assessment and education may not lead to reduction in suicides, and implementation is often incomplete or ineffective. The actions may also have a function in assuaging anxieties and risk *for the system* while they ostensibly also seek to reduce suicide per se.

You absolutely have an important point here.

The discussions should include reference to the work of the UK National Confidential Inquiry into Suicide and Homicide, which was able to draw conclusions about the effectiveness of suicide prevention actions, eg:

Kapur N, Ibrahim S, While D, Baird A, Rodway C, Hunt IM, et al. Mental health service changes, organisational factors, and patient suicide in England in 1997–2012: a before-and-after study. *Lancet Psychiatry* 2016;3: 526–34

And also the impact of hospitalisation on risks:

Kapur N, Steeg S, Turnbull P, Webb R, Bergen H, Hawton K, et al. Hospital management of suicidal behaviour and subsequent mortality: a prospective cohort study. *Lancet Psychiatry* 2015;2: 809–16.

Thank you for these excellent references! We have read and found the paper from 2016 relevant as a reference in the paper (p 4: 4-6, ref 28).

I think that organisational psychology and dynamics have a significant influence on what is measured, reported and changed (or not changed). Although beyond the scope of this paper, it would be useful to see that aspect of risk management acknowledged.

I think this is a thorough and worthwhile paper and would support publication, subject to the observations above. The limitations acknowledged by the authors and which I’ve referenced above are not only for this paper, but for this field of enquiry as a whole.

I’d therefore politely suggest that the paper would be enhanced by an accompanying editorial setting out the need to critically examine the purpose of these reports, and to make the case for extending the “frame” for this work to include a much broader set of influences.

VERSION 2 – REVIEW

REVIEWER	Fredrik A. Walby, PsyD National Center for Suicide Research and Prevention University of Oslo, Norway
REVIEW RETURNED	11-Feb-2021

GENERAL COMMENTS	In my opinion, the authors have done a thoughtful job revising the manuscript, and I have just one small comment: The STROBE checklist is rightfully used and attached to the submission, but I miss a short mention of compliance with the guideline in the methods section.
---

VERSION 2 – AUTHOR RESPONSE

*Thank you so much for your helpful and wise comments!
We have made revisions in the paper in concordance with your suggestions and our comments below.*

Please revise your title so that it includes your study design. This is the preferred format for the journal

Thank you, we revised the title, p 1.

- Please revise the abstract so that it is following the structured abstract recommended in BMJ Open's Instructions for Authors for research articles. See: <https://bmjopen.bmj.com/pages/authors/#research>
The reporting of the methods in the abstract needs to be more detailed whilst the conclusions section needs to be a lot shorter

Thank you, we now have revised the abstract so it follows the recommended structure and made revisions as you suggested.

- Please ensure that you refer to the STROBE checklist from within the Methods, as the reviewer suggests.

Of course, we added this to the method section, p4, line 42-43.

- Under Data availability and within the Methods section, please provide more details of the databases used in your study. Please specify whether you obtained permission to use these data and who granted it, or whether the data are publicly available. If not freely available, please specify how they can be requested and from whom.

Thank you for this point. We added this information in the method section, p5, line 10, and under Data availability, p 14, line 20-23.

Reviewer: 1

Mr. Fredrik Walby , University of Oslo

Comments to the Author:

In my opinion, the authors have done a thoughtful job revising the manuscript, and I have just one small comment: The STROBE checklist is rightfully used and attached to the submission, but I miss a short mention of compliance with the guideline in the methods section.

Thank you! We added the information about the STROBE checklist to the method section, p4, line 42-43.